# Diagnostic and treatment delay in extrapulmonary tuberculosis and association with mortality: Experiences from Mbeya, Tanzania

Erlend Grønningen[1,2]*, Marywinnie Nanyaro[3], Esther Ngadaya[3], Tehmina Mustafa[1,2]

**1** Department of Global Public Health and Primary Care, Centre for International Health, University of Bergen, Bergen, Norway, **2** Department of Thoracic Medicine, Haukeland University Hospital, Bergen, Norway, **3** National Institute for Medical Research, Muhimbili Medical Research Centre, Dar es Salaam, The United Republic of Tanzania

* egr025@gmail.com

## Abstract

### Background

Unlike pulmonary tuberculosis, there is limited information on delays in diagnosis and treatment initiation in extrapulmonary tuberculosis (EPTB) and their consequences for disease outcomes and mortality. In low- and middle-income countries, most EPTB cases are presumed rather than microbiologically confirmed, which might lead to an underestimation of the mortality rates in EPTB.

### Objective

The study aimed to assess the delays in diagnosis and treatment in EPTB and their association with mortality in a setting with a high prevalence of both HIV and malnutrition.

### Method

We included 106 EPTB patients from Mbeya Zonal Referral Hospital, who were followed up until the completion of their treatment. Patients were classified as having EPTB using a clinical case definition. In total, 37 of 106 (35%) EPTB cases resulted in death. The median (interquartile range) total diagnostic delay for survivors was 59 days (26-136), while for those who died, it was 78 days (32-165). The corresponding median (interquartile range) treatment delay was 66 days (33-140) for survivors and 78 days (27-189) for those who died. None of the differences reached statistical significance when analyzed with non-parametric tests. Surprisingly, 21 patients did not receive TB treatment, but this lack of therapy did not affect mortality or correlate with a longer diagnostic delay.

### Conclusion

We were unable to demonstrate that diagnostic or treatment delays were higher in EPTB patients who died. Furthermore, EPTB patients who did not receive TB treatment did not

**Data availability statement:** All data analyzed in this article are submitted with the article and the full dataset is available in the supporting information.

**Funding:** Funding was provided by Norges Forskningsråd (234457), European and Developing Countries Clinical Trials Partnership (234457) and also by Helse Vest (234457).

exhibit higher mortality rates. Further prospective studies with larger sample sizes are needed to better understand the factors contributing to delays in diagnosis and treatment, as well as their potential impact on mortality in EPTB.

## Introduction

According to the World Health Organization (WHO), tuberculosis (TB) probably returned to being the leading cause of infectious disease-related deaths globally in 2023. It is estimated that 10.8 million fell ill with the disease, resulting in an estimated 1.3 million deaths in 2023 [1].

Extrapulmonary tuberculosis (EPTB) accounted for 16% of notified TB cases worldwide in 2023 and has been identified as a risk factor for delayed TB diagnosis [2–5]. People living with HIV (PLHIV), particularly those with lower CD4 counts, as well as children, have a higher prevalence of EPTB [6, 7]. Extensive studies have analyzed factors associated with delayed diagnosis in pulmonary TB (PTB) due to its infectiousness, in order to ensure that TB programs detect and treat this disease, which poses a threat to public health [2,8–10]. However, less is known about delays in diagnosis and treatment in EPTB, as well as the consequences of these delays for individuals and health systems [3]. Extrapulmonary tuberculosis patients are often perceived as a lesser public health threat, as they are rarely infectious [11]. Few recent studies have analyzed healthcare-seeking behavior, diagnostic and treatment delay in presumptive EPTB cases, though they did not assess the correlation with mortality [11–13].

Research on delays in diagnosis and treatment in EPTB can be challenging due to several factors. These include the need for and cost of invasive sampling of EPTB lesions, the use of imperfect diagnostic tools, the infrequent application of these tools, a lack of uniform case definitions, the observation that most notified EPTB cases are diagnosed clinically in the low-resource setting [14–17]. The consequences of delayed initiation of anti-tuberculosis treatment (ATT) can be severe in conditions like TB meningitis and pericarditis, but less is known about the effects of delayed diagnosis and ATT initiation on mortality for other sites of infection [18, 19]. There is no consensus on what constitutes an acceptable time to diagnosis or treatment for PTB or EPTB, as this interpretation depends on the available healthcare infrastructure. In individual studies, the median time is often used at the cut-off for what is considered an acceptable delay in diagnosis or treatment [10,20].

The United Republic of Tanzania is one of the 30 high burden TB countries, with a concurrent HIV epidemic. Since 2015, Tanzania has reduced TB-related deaths by more than 50% [1]. In 2023, the WHO estimated 122,000 TB cases nationally, of which 92,000 were notified, 17% of which were HIV infected. EPTB accounted for 16% of all TB cases. The estimated TB case fatality rate was 17% in 2023, and 49% of PTB cases were bacteriologically confirmed. In comparison, only 206 (1.2%) EPTB cases were bacteriologically confirmed in Tanzania in 2021 [21].

In our previous research at Mbeya Zonal Referral Hospital (MZRH), conducted to validate the MPT64 test as a new diagnostic tool for EPTB, we published all-cause mortality rates of 23% in children and 40% in adults with EPTB [14,22]. Being hospitalized was identified as the primary risk factor for mortality. The aim of this study was to assess whether delays in diagnosis and treatment in EPTB contributed to the high mortality rates observed in both children and adults.

## Methods

### Study design

From April 2016 to July 2017, we conducted a prospective cohort study to validate the MPT64 test for EPTB in MZRH [14, 15]. The EPTB cases analyzed in our previous studies

on validation of the MPT64 test and mortality rates were pooled in this paper to analyze the effects of delays in diagnosis and treatment on mortality in EPTB [14,15,22]. The enrollment of patients and implementation of the MTP64 test were carried out regardless of sample size.

## Health system in the region

MZRH is a tertiary-level public hospital under the Ministry of Health, serving the Southern Highlands zone of Tanzania, which includes the regions of Iringa, Njombe, Ruvuma and the Southwest Highlands zone (Mbeya, Rukwa, Katavi, Songwe). Referrals follow a stepwise process in each region, beginning with primary health care services such as dispensaries and health centers at the divisional level. Patients are then referred to district hospitals, regional hospitals, and, finally to the zonal referral hospitals. Additionally, private hospitals provide services ranging from primary care to specialized consultations. Tanzania operates a decentralized health system.

## Patient enrollment

Both the in- and outpatient services at MZRH were involved in patient enrollment. Clinicians were asked to include patients of all ages with a clinical suspicion of EPTB. Samples were collected from the site of infection in all patients and subjected to routine diagnostic tests, including mycobacterial culture, the Xpert MTB/RIF assay (Cepheid, Sunnyvale, United States of America) (Xpert), acid fast bacilli staining, and cytology, as previously published [14,15,22]. Cytology, acid fast bacilli staining, and the immunochemistry-based MPT64 test were analyzed locally at the MZRH histopathology unit. Samples for mycobacterial culture and Xpert testing were sent to the Central Tuberculosis Reference Laboratory (CTRL) at Muhimbili National Hospital in Dar es Salaam. ATT was initiated by the treating physicians.

## Exclusion criteria

Patients lost to follow up were not analyzed in this study. Patients with missing data on diagnostic or treatment delay were excluded from the study. For the analysis on treatment delay, patients already on ATT at the time of inclusion were excluded, and only those initiated on ATT during the study were included in the analysis.

## Follow-ups

Follow-up visits were arranged to assess the response to ATT at 2-3 months and at the end of treatment. For patients initially lost to follow-up, additional tracing was performed in January 2018. After the last patient was included in July 2017, the follow-up continued for an additional six months for the mortality analysis with data collection concluded in January 2018. Response to treatment was defined as a $\geq 3$-point improvement in symptoms as measured by patient (I) or clinician (I), weight gain (I) and improvement of objective findings such as size of lymph nodes, pleural effusions and ascites (I) [14,15,23]. Response to treatment was assessed at both 2-3 months and 6 months.

## Data sources

Upon inclusion, patients completed a questionnaire regarding their prior medical history, healthcare-seeking behavior, onset of symptoms and treatment received. The researchers were granted access to the electronic hospital records of MZRH and Baylor College of Medicine Children´s Foundation Tanzania (Baylor) as part of the study. Baylor provides comprehensive pediatric care and treatment for conditions such as TB, HIV/AIDS, malnutrition, and other complex conditions in the region.

## Definitions

### Presumptive EPTB patients

In our previous papers, patients were classified into TB and non-TB groups [14, 15]. In this study we analyzed the TB group. Patients in our previous studies were categorized as either bacteriologically confirmed or clinically diagnosed TB cases (probable and possible) using a composite reference standard that graded the certainty of the diagnosis [14–16,22]. A bacteriologically confirmed case was positive on mycobacterial culture/Xpert. A probable case was a clinically presumed EPTB that responded to ATT at 2-3 months or the end of treatment, or a clinically presumed EPTB case with confirmed pulmonary TB, accompanied by a positive objective finding on either radiological examinations, cytology, or fine needle aspiration. A possible case had strong clinical suspicion of TB and responded to treatment, or showed findings on radiology, or TB-specific findings on cytology/fine needle aspiration. The three groups were merged into a single "TB" category in the current analysis.

### Delays

Patient delay was defined as the self-reported time from onset of symptoms to the first contact with any healthcare provider [2,10]. The onset of symptoms was reported in the study questionnaire. Healthcare system diagnostic delay was defined as the time from first contact with a healthcare provider to inclusion in the study for diagnostic sampling of suspected EPTB lesion(s) [2,10]. Information regarding the first healthcare contact was reported in the study questionnaire, while the date of inclusion in the study and sampling was reported in both the study questionnaire and the electronic hospital records. Total diagnostic delay was the sum of patient delay and healthcare system diagnostic delay [2,10]. Delay in ATT initiation was the time from diagnostic sampling to initiation of ATT. Total treatment delay was the time from onset of symptom to initiation of ATT. The date of ATT initiation was obtained from the TB treatment card and the electronic hospital records. The different delays are visualized in Fig 1.

### Mortality

Patients were classified as deceased if death was certified in the electronic hospital records or if the information was provided during phone follow-up. Death from any cause was considered as mortality. Patients who responded on the phone follow-ups or had continuous notes in the hospital records until the end of the study were considered alive.

### Ethical statement

After receiving information about the study, all included patients gave their informed written consent. Guardians gave consent for pediatric patients. Patients were asked to disclose their HIV serostatus, and additional information was retrieved in the hospital records. The decision to request HIV testing was made by the clinicians in MZRH according to national guidelines. A review of the hospital records of the included patients was approved. Ethical approval for the study was received from both the regional Committee for Medical Research Ethics in Norway, REK Helse-Vest (2014/46/REK vest) and the Ethical committee for biomedical research at the National Medical Research Coordinating Committee in Tanzania (NIMR/HQ/R.8a/Vol. IX/2142). All methods were performed in accordance with relevant guidelines and regulations.

### Statistical analysis

For statistical analysis, Statistical Package for Social Sciences, version 28.0 (IBM, Armonk, NY), was used. To analyze factors associated with mortality, odds ratios (OR) for categorical variables were calculated using logistic regression. ORs were considered statistically significant at the 5%

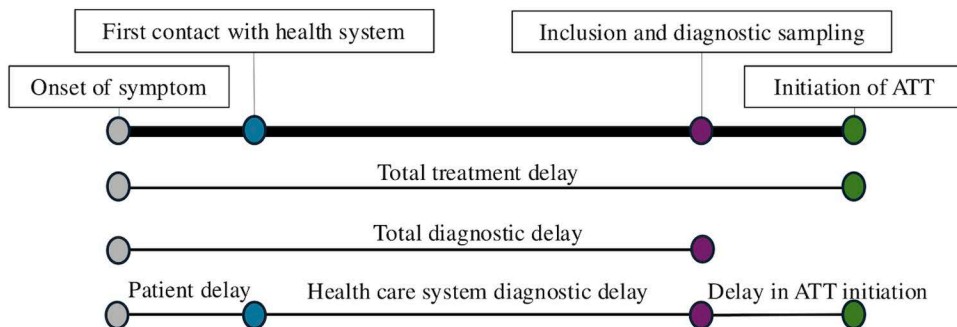

**Fig 1. Illustration of different delays experienced by patients. Patient delay**: The time (days) from patient´s report of symptom onset to the first visit to a healthcare provider. **Healthcare system diagnostic delay**: The time from the first healthcare visit to diagnostic sampling for suspected EPTB. **Total diagnostic delay**: The sum of patient delay and healthcare system diagnostic delay. **Delay in ATT initiation**: The time from diagnostic sampling to the start of ATT. **Total treatment delay**: The time from symptom onset to the start of ATT.

level if the 95% confidence intervals (CI) did not include 1.0. Adjusted odds ratios (aOR) were calculated by including all the variables that were significant in a multinominal logistic regression model, and statistical significance was defined in the same way as for ORs. For analyses on mortality and diagnostic/treatment delays, Mann-Whitney-test was applied for group comparisons, as the data were non-normally distributed. A p-value < 0.05 was considered statistically significant.

## Results

### Diagnostic and treatment delay

A total of 106 EPTB patients were included in the study, of whom 37/106 (35%) had died by the time of follow-up. The patient delay, health system diagnostic delay, total diagnostic delay, and total treatment delay in days is visualized in the box plots in Fig 2. Healthcare system diagnostic delay was longer than patient delay. No significant difference was found in any of the delays between surviving and deceased EPTB patients.

The median (interquartile range, IQR) total diagnostic delay for patients who survived was 59 days (26-136), while for those who died, the corresponding delay was 78 days (32-165). For treatment delay, the corresponding figures were 66 days (33-140) for survivors and 78 days (27-189) for those who died. The delay in ATT initiation from diagnostic sampling of suspected infection site(s) was short: 7 days (1–11) for survivors versus 4 days (0–9) for the deceased (p = 0.14).

For total diagnostic delay in adenitis, the median was 60 days (22-133) for survivors versus 62 days (29-183) for those who died. In non-adenitis cases, the delay was 43 days (26-155) for survivors versus 80 days (32-162) for those who died, with no statistically significant differences between the groups.

The boxes represent the interquartile range (IQR) and median, while the whiskers extend to 1.5 x IQR. Outliers were removed from the visualization as they spanned a large range of values. In the group of survivors, the maximum value was 1193 days, and in the deceased group 375 days. No significant differences observed (p-value for survivors vs deceased for respective delays)

a. Delays in all survivors (N = 69) vs deceased (N = 37) EPTB patients. ATT delay is shown for all survivors (N = 55) vs deceased (N = 16) EPTB patients who were started on ATT after inclusion in the study. Patients who were on ATT on inclusion were excluded.

b. Delays in survivors (N=36) vs deceased (N=6) adenitis patients.

c. Delays in survivors (N = 33) vs deceased (N = 31) non-adenitis patients.

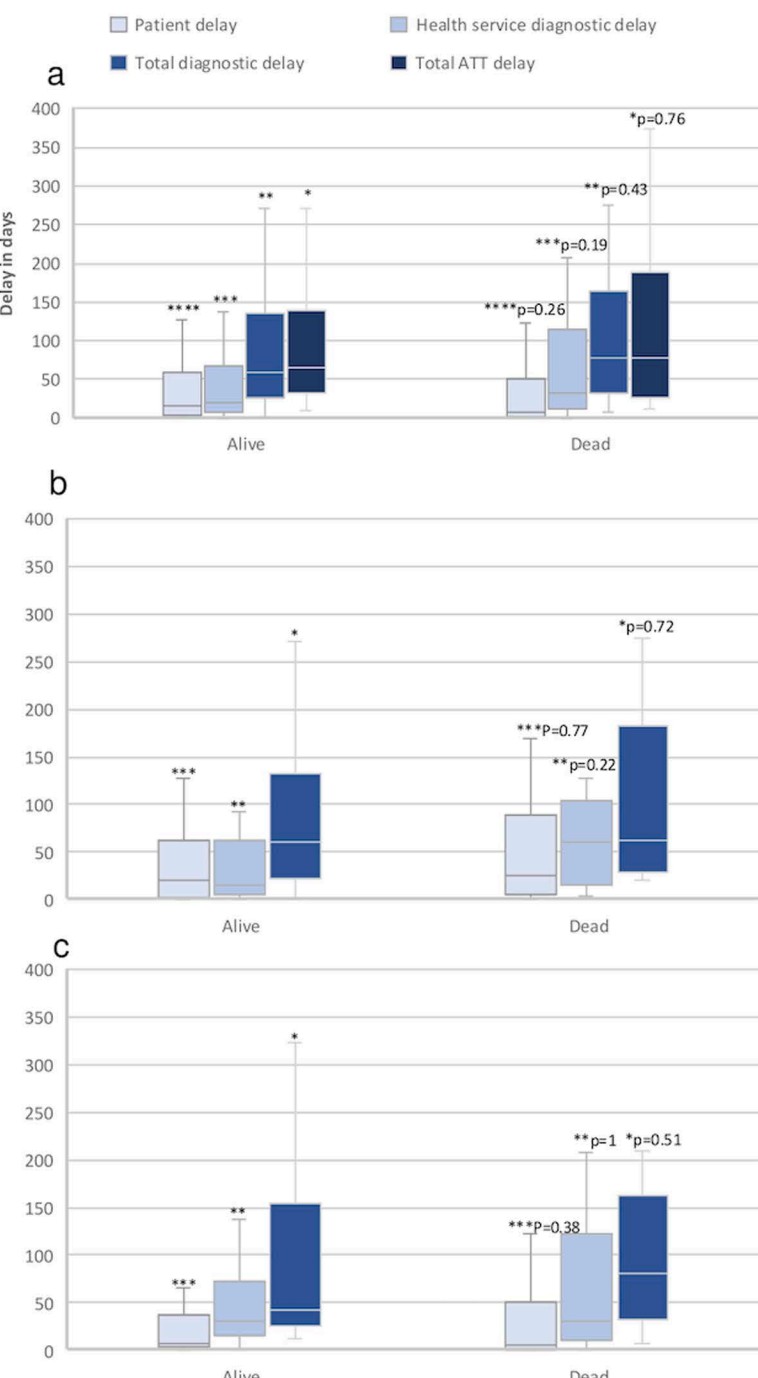

**Fig 2. Box plot of patient delay, healthcare system diagnostic delay, total diagnostic delay, and total delay in anti-tuberculosis therapy (ATT).**

## Characteristics of patients in the study

Most patients in the study were male, but gender did not affect mortality. The oldest age group (>35 years) had significantly higher odds of dying compared to the youngest age group (0-14 years), with an aOR of 6.20 (95% CI: 1.46-26.40), as shown in Table 1.

**Table 1. Characteristics of extrapulmonary tuberculosis patients according to outcomes.**

| | All patients (N = 106) | | | |
|---|---|---|---|---|
| | Died (N = 37, 35%) | Survived (N = 69, 65%) | OR (95% CI) [1] | aOR (95% CI) [1] |
| Gender | | | | |
| Female | 9/37 (24%) | 32/69 (46%) | 1 | – |
| Male | 28/37 (76%) | 37/69 (64%) | 2.69 (1.11-6.54) | 2.52 (.83-7.65) |
| Age group | | | | |
| 0-14 years | 5/37 (14%) | 18/69 (26%) | 1 | – |
| 15-35 years | 12/37 (32%) | 31/69 (45%) | 1.29 (.42-4.60) | – |
| >35 years | 20/37 (54%) | 20/69 (29%) | 3.60 (1.20-11.58) | 6.20 (1.46-26.40) |
| Hospitalized at any point? | | | | |
| Hospitalized | 36/37 (97%) | 36/69 (52%) | 33.00 (4.28-254.39) | 34.66 (3.13-383.55) |
| Outpatient | 1/37 (3%) | 33/69 (48%) | 1 | – |
| HIV status | | | | |
| Uninfected | 17/37 (46%) | 35/69 (51%) | 1 | – |
| Unknown | 2/37 (5%) | 4/69 (6%) | 1.03 (.17-6.19) | – |
| Infected | 18/37 (49%) | 30/69 (43) | 1.24 (.54-2.81) | – |
| Antiretroviral therapy (ART) coverage (N = 48) | | | | |
| On ART | 16/18 (89%) | 29/30 (97%) | 1 | – |
| Not on ART | 2/18 (11%) | 1/30 (3%) | 3.63 (.31-43.15) | – |
| On TB treatment or TB treatment started | | | | |
| Yes | 22/37 (60%) | 63/69 (91%) | 1 | – |
| No | 15/37 (40%) | 6/69 (9%) | 7.16 (2.47 -20.75) | 3.20 (.86-11.93) |
| Percentage responding to TB treatment (N = 82) [2] | 0/22 (0%) | 54/60 (90%) | – | |
| Malnutrition in 0-14 age group (N = 23) [3] | | | | |
| Yes | 3/5 (60%) | 9/18 (50%) | 1.50 (.20-11.24) | – |
| No | 2/5 (40%) | 9/18 (50%) | 1 | – |
| Any comorbidity present [4] | | | | |
| Yes | 32/37 (86%) | 47/69 (68%) | 3.00 (1.03-8.73) | 1.68 (.40-7.10) |
| No | 5/37 (14%) | 22/69 (32%) | 1 | – |
| Site of infection | | | | |
| Adenitis | 6/37 (16%) | 36/69 (52%) | 1 | – |
| All other EPTB forms | 31/37 (84%) | 33/69 (48%) | 5.64 (2.09-15.22) | .95 (.24-3.81) |
| Pleuritis | 9/37 (24%) | 9/69 (13%) | 6.00 (1.69-21.26) | – |
| Peritonitis | 8/37 (22%) | 8/69 (11%) | 6.00 (1.62-22.16) | – |
| Meningitis | 3/37 (8%) | 3/69 (4%) | 6.00 (.97-36.97) | – |
| PTB with EPTB [5] | 4/37 (11%) | 6/69 (9%) | 4.00 (.86-18.51) | – |
| Multiorgan [6] | 5/37 (14%) | 6/69 (9%) | 5.00 (1.15-21.70) | – |
| Other [7] | 2/37 (5%) | 1/69 (1%) | 12.00 (.94-153.89) | – |

[1]Odds ratio calculation by logistic regression. Group with lowest odds of dying given value 1 and used as reference. Significant factors analyzed by logistic regression (aOR) model.

[2]Three cases with unknown response to treatment removed.

[3]Malnutrition includes Severe Acute Malnutrition and Moderate Acute Malnutrition

[4]For grouping of comorbidities see supplementary file 1.

[5]Pulmonary tuberculosis with extrapulmonary manifestation

[6]Includes 2 cases of disseminated TB, 8 cases with EPTB two sites and 1 miliary TB.

[7]Includes 2 cysts in abdomen and 1 mastitis.

The majority of patients included were hospitalized, and hospitalization was identified as the major risk factor for mortality (aOR 34.66, 95% CI: 3.13-383.55). No significant differences were found in mortality based on HIV serostatus. In total, 48/106 (45%) were HIV infected. Antiretroviral therapy (ART) coverage was high in both the deceased 16/18 (89%) and surviving 29/30 (97%) groups.

Most surviving patients received ATT, 63/69 (91%). However, in the group that died, only 22/37 (60%) received ATT, and 15/37 (40%) were not treated with ATT. Despite this, when adjusting for other factors, the lack of ATT was not significantly associated with mortality (aOR 3.20, 95% CI:.86-11.93). Among patients who survived and received ATT, 57/63 (90%) had a good response to the treatment.

Comorbidities were common in both groups, but there were no significant differences between them. Malnutrition was the most prevalent comorbidity in the youngest age group 12/23 (52%), but it did not differ according to outcome.

As shown in Table 1, pleuritis 9/37 (24%), peritonitis 8/37 (22%), and adenitis 6/37 (16%), were the three most common sites of infection associated with death. However, we were unable to demonstrate a significantly higher adjusted odds of dying in non-adenitis cases compared to adenitis cases.

## Health facilities visited

Table 2 presents a comparison of the health facilities visited by patients who survived versus those who died. Nearly half of the patients in both groups had visited only one health facility for their current problem (18/37 (49%) in the deceased group vs 45/66 (68%) in the surviving group). Visiting multiple health facilities was not associated with a higher odds of death.

The majority of patients in both groups had either one (10/37 (27%) vs 20/68 (29%)) or two visits (15/37 (41%) vs 24/68 (35%)) to a health facility for their current symptom. No association with mortality was found based on the number of visits. Most patients in both groups sought care in a hospital (30/37 (81%) vs 47/68 (69%)). As shown in Table 2, self-referral and referral from a hospital or dispensary were the main pathways of referral in both groups.

## Diagnostic delay and characteristics of patients not receiving ATT

For the subgroup of 21 patients who did not receive ATT, the median (IQR) of the total diagnostic delay was 115 days (39-168), compared to 61 days (26-136) for those who received ATT. However, the difference did not reach statistical significance (p = 0.087).

Descriptive data for the 21 patients who did not receive ATT is shown in Table 3. Five of these patients were positive on mycobacterial culture and were classified as confirmed TB cases according to our clinical case definition, while the remaining cases were considered possible cases. Only 5/21 (24%) had no comorbidities. Three patients had received ATT within the last 12 months, two of whom died.

Of the 15 patients who died, 7/15 (47%) died within the first four days of inclusion in the study, all while hospitalized. Seven of the remaining eight had a longer time to death (16 days or more after inclusion), and five of these seven (71%) were positive on at least one of the following tests: mycobacterial culture, Xpert, or MPT64 test, but were not started on ATT.

## Discussion

The high mortality rates observed in our study (35% of EPTB patients) are not widely described in the literature, although they have been reported in recent studies [22,24,25]. Diagnostic and treatment delays were not significantly associated with mortality. A multivariate logistic regression model identified hospitalization and older age (>35 years old) as

**Table 2. Comparison of health facilities visited by extrapulmonary tuberculosis patients who survived or had died on follow-up.**

| | | TB patients (N = 106) | | | |
| --- | --- | --- | --- | --- | --- |
| | | Died (N = 37, 35%) | Survived (N = 69, 65%) | OR¹ (95% C.I.) | aOR¹ (95% C.I.) |
| Number of health facilities visited (N = 103) | One | 18/37 (49%) | 45/66 (68%) | 1 | – |
| | Two | 13/37 (35%) | 15/66 (23%) | 2.17 (.86-5.45) | – |
| | Three or more | 6/37 (16%) | 6/66 (9%) | 2.50 (.71-8.78) | – |
| Number of visits to health facility (N = 105) | First visit | 10/37 (27%) | 20/68 (29%) | 1 | – |
| | Second visit | 15/37 (41%) | 24/68 (35%) | 1.25 (.46-3.39) | – |
| | Third visit | 5/37 (13%) | 10/68 (15%) | 1.00 (.27-3.72) | – |
| | >Three visits | 7/37 (19%) | 14/68 (21%) | 1.00 (.31-3.26) | – |
| Referral to this facility (N = 102) | Self-referral | 9/35 (26%) | 31/67 (46%) | 1 | – |
| | Government dispensary/health center/hospital | 15/35 (43%) | 23/67 (34%) | 2.25 (.84-6.03) | – |
| | Private hospital/dispensary | 2/35 (5%) | 6/67 (9%) | 1.15 (.20-6.70) | – |
| | Family/NGO/religious leader/healer/pharmacy/village health worker | 9/35 (26%) | 7/67 (11%) | 4.43 (1.29-15.23) | .34 (.06-1.91) |
| First health facility visited (N = 105) | Hospital (private, district, regional, referral) | 30/37 (81%) | 47/68 (69%) | 1 | – |
| | Health center/dispensary | 5/37 (13%) | 17/68 (25%) | .46 (.15-1.38) | – |
| | Pharmacy | 1/37 (3%) | 2/68 (3%) | .78 (.07-9.02) | – |
| | Healer | 1/37 (3%) | 2/68 (3%) | .78 (.07-9.02) | – |

¹Odds ratio calculated using logistic regression. Reference group given value 1. Factors that were significant included in logistic regression model and adjusted OR calculated.

key factors associated with increased odds of dying. HIV serostatus, treatment with ATT and comorbidities did not significantly differ between survivors or deceased.

The health system delay contributed more than patient delay towards the total diagnostic delay in our study. Previous studies report longer diagnostic delays in EPTB compared to PTB [2,5]. While the diagnostic delays observed in our surviving patients align well with those reported from Zanzibar, the delays for those who died were longer than published previously [11,13].

Twenty-one EPTB patients who did not receive ATT had a median total diagnostic delay of 115 days (IQR 39-168), compared to 61 days (IQR 26-136) for those who received treatment. Though this difference did not reach statistical significance (p = 0.087), the mortality rate was high in this group (71%), with half dying within 4 days of hospitalization. It is unlikely that starting ATT based on clinical suspicion on admission would have changed the outcomes for these patients. Two of these cases (meningitis and adenitis) had a shorter diagnostic delay of 21 and 32 days, but all the other cases had a diagnostic delay above 50 days, showing that an earlier diagnosis and ATT potentially could have saved them. Furthermore, 7/15 (47%) died after 16 days or more after inclusion in the study, underscoring that these patients had a significant delay in appropriate care through ATT that possibly could have reduced mortality. Notably, a positive confirmatory diagnostic test was found in 5/7 of these cases, yet they did not receive ATT. Linking patients to treatment after diagnostic sampling has also been shown to be a problem for PTB [26], and underlines the need for systems strengthening in parallel with implementation of new

**Table 3. Characteristics of 21 patients who did not receive anti tuberculosis therapy (ATT).**

| No. | Hospi-talized | Dies | Age | Site | HIV/ART [1] | Comor-bidity [2] | Cul-ture [3] | Xpert | MPT64 | ATT <12M? | TB group | Cyt. [4] | FNAC [5] | Rad. [6] | D1 [7] | D2 [7] | D3 [7] |
|---|---|---|---|---|---|---|---|---|---|---|---|---|---|---|---|---|---|
| 1 | Yes | Yes | 57 | Multiorgan | -/NA | CHF | + | – | + | No | Confirmed | + | NA | – | 120 | 34 | 154 |
| 2 | Yes | Yes | 20 | Adenitis | -/NA | None | + | + | + | No | Confirmed | NA | + | + | 60 | 45 | 105 |
| 3 | Yes | Yes | 68 | Pleuritis | M/NA | None | M | M | + | No | Possible | + | NA | + | 75 | 1 | 76 |
| 4 | Yes | No | 48 | Pleuritis | -/NA | None | + | + | + | No | Confirmed | + | NA | + | 23 | NA | NA |
| 5 | No | No | 43 | Adenitis | -/NA | None | – | – | + | No | Possible | NA | + | + | 128 | NA | NA |
| 6 | No | No | 28 | Adenitis | -/NA | None | – | – | + | No | Possible | NA | + | – | 433 | NA | NA |
| 7 | Yes | Yes | 69 | Peritonitis | -/NA | Anemia | – | – | + | No | Possible | – | NA | + | 162 | 143 | 305 |
| 8 | Yes | No | 30 | Pleuritis | +/+ | HIV | – | – | M | No | Possible | + | NA | + | 38 | NA | NA |
| 9 | Yes | Yes | 39 | Peritonitis | -/NA | RF + LC | – | – | + | No | Possible | + | NA | + | 561 | M | M |
| 10 | Yes | Yes | 42 | Peritonitis | +/- | Anemia | M | – | – | No | Possible | + | NA | + | 59 | 3 | 62 |
| 11 | Yes | Yes | 73 | Peritonitis | -/NA | RF + LC | – | – | + | No | Possible | + | NA | + | 174 | 4 | 178 |
| 12 | Yes | Yes | 30 | Adenitis | +/+ | RF | + | + | M | Yes | Confirmed | NA | + | – | 275 | 52 | 327 |
| 13 | Yes | No | 54 | Pleuritis | +/+ | Syphilis | – | – | – | Yes | Possible | + | NA | + | 323 | NA | NA |
| 14 | Yes | No | 75 | PTB w/ EPTB | -/NA | CT | – | – | M | No | Possible | + | NA | + | 39 | NA | NA |
| 15 | Yes | Yes | 36 | Pleuritis | +/+ | Empyema | – | – | – | No | Possible | + | NA | + | 32 | 157 | 189 |
| 16 | Yes | Yes | 47 | Peritonitis | +/+ | RF | – | – | – | No | Possible | + | NA | + | 115 | 1 | 116 |
| 17 | Yes | Yes | 24 | Cyst in abdomen | -/NA | Schisto. | – | – | + | No | Possible | NA | + | + | 158 | 16 | 174 |
| 18 | Yes | Yes | 48 | Adenitis | +/+ | CrM | – | – | M | No | Possible | NA | + | – | 21 | 1 | 22 |
| 19 | Yes | Yes | 36 | PTB w/ EPTB | +/+ | KS | – | – | – | Yes | Possible | + | NA | + | 124 | 34 | 158 |
| 20 | Yes | Yes | 9 | Meningitis | -/NA | SAM | – | – | M | No | Possible | + | NA | NA | 32 | 0 | 32 |
| 21 | Yes | Yes | 9 | Meningitis | -/NA | SAM | + | + | M | No | Confirmed | – | NA | NA | 104 | 0 | 104 |

[1] NA = Not applicable, + = HIV infected, on ART, - = HIV negative, not on ART

[2] Major comorbidity CHF = congestive heart failure, RF = Renal failure, LC = Liver cirrhosis, CT = Cardiac tamponade, Schisto = Schistosomiasis, CrM = Cryptococcal meningitis, KS=Kaposi's sarcoma, SAM = Severe acute malnutrition

[3] M = Missing, + = positive test result, - = negative test result

[4] Positive finding on cytology: Finding of predominantly lymphocytes and/or macrophages, or epithelioid cells and/or necrosis

[5] Positive finding on FNAC: Finding of granulomatous inflammation with/without necrosis or any necrosis (excluding suppurative inflammation)

[6] Positive finding on radiology: Pleural effusion and/or infiltrates/nodules consistent with tuberculosis noted on chest x-ray, signs of skeletal TB present or CT findings compatible with TB.

[7] D1 = Diagnostic delay in days, D2 = Days to death from inclusion, D3 = Days to death from symptom

diagnostic tools. Overtreatment of presumed EPTB cases is well described in the literature [14–17,27], but the effect of not treating EPTB is not well described besides for meningitis and pericarditis [19,28].

Most cases who did not receive ATT were classified as "possible TB" cases (16/21). Our case definition was applied post hoc, meaning clinicians may not have considered TB initially when starting treatment [14, 15]. This gap in clinical decision-making may have contributed to the lack of treatment.

Comorbidities were prevalent in both groups and might help explain the high case fatality rates observed in sites such as pleuritis (9/18, 50%) and peritonitis (8/16, 50%) where mortality rates are not normally reported to be this high. The unexpectedly high mortality rates among TB adenitis cases is likely due to misclassification of hospitalized

TB adenitis patients that had multiorgan TB [22]. At the time of the study no CT scan was available in MZRH.

The type of health facilities visited, or number of visits did not influence mortality. Previous studies have described a vicious cycle of repeated visits to the same healthcare level and a lack of referrals [2]. However, our study, did not find evidence of this; most patients had visited one or two clinics once or twice, and self-referral or referral from a hospital were the primary pathways, indicating a functional referral systems in the region.

The data in this paper comes from a real-life study where implementation of a new diagnostic test was evaluated in the routine care. However, it is likely that the intervention introduced a Hawthorne effect [29], and that the clinicians requested more diagnostic sampling of suspected EPTB sites than is common in regular clinical practice. This is highlighted by the fact that the yield of EPTB mycobacterial cultures in our study approximates the annual yield on EPTB samples for all of Tanzania [30]. Moreover, since sampling was free of charge in our study, diagnostic delays in regular clinical settings may be longer than reported here.

We included sample processing time in the analysis of ATT delay. The processing time potentially could have influenced delay in ATT initiation if clinicians had chosen to withhold ATT until they had a confirmed EPTB diagnosis.

Limitations of this study include a small sample size, merging data from different EPTB sites (including meningitis), and analyzing both inpatient and outpatient cases together. Patient delay is subject to both recall and reporting bias, and removing patients that were lost-to-follow-up from the mortality analysis may have both overestimated and underestimated mortality, as other studies have found that such patients often have higher mortality rates [25].

## Conclusion

We were unable to show that diagnostic delay or treatment delay affected the high mortality rates observed in our cohort of EPTB patients. Further prospective studies with larger sample sizes are needed to better understand the causes of diagnostic delay and their impact on mortality.

## Supporting information

**S1 File. Classification of comorbidities.**
(PDF)

**S2 File. Dataset**
(XLSX)

## Acknowledgments

We thank the National Tuberculosis and Leprosy Programme and the Central Tubercuosis Reference Laboratory Laboratory at Muhimbili National Hospital in Dar es Salaam, for their support and analysis of samples. For their dedication to the patients in the study and the recruitment of participants we thank the clinicians in Mbeya Zonal Referral Hospital. Anthony Ambikile Nsojo in the Research department at Mbeya Zonal Referral Hospital and dr Lisete Torres and dr William Muller from the Histopathology unit at Mbeya Zonal Referral Hospital were key people in making this study happen and we thank them sincerely. For the development of the questionnaire used in the study the authors thank Melissa Jørstad. We thank Baylor for their follow up of the pediatric patients and their collaboration.

## Author contributions

**Conceptualization:** Erlend Grønningen, Esther Ngadaya, Tehmina Mustafa.

**Data curation:** Erlend Grønningen, Marywinnie Nanyaro.

**Formal analysis:** Erlend Grønningen, Tehmina Mustafa.

**Funding acquisition:** Tehmina Mustafa.

**Investigation:** Erlend Grønningen.

**Methodology:** Erlend Grønningen, Tehmina Mustafa.

**Project administration:** Marywinnie Nanyaro, Esther Ngadaya, Tehmina Mustafa.

**Resources:** Esther Ngadaya, Tehmina Mustafa.

**Supervision:** Marywinnie Nanyaro, Tehmina Mustafa.

**Writing – original draft:** Erlend Grønningen.

**Writing – review & editing:** Erlend Grønningen, Marywinnie Nanyaro, Esther Ngadaya, Tehmina Mustafa.

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
