## [Decision Letter · Decision Letter 0]

25 Nov 2024

PONE-D-24-41844Diagnostic and treatment delay in extrapulmonary tuberculosis and association with mortality: experiences from Mbeya, Tanzania.PLOS ONE

Dear Dr. Grønningen,

Thank you for submitting your manuscript to PLOS ONE. After careful consideration, we feel that it has merit but does not fully meet PLOS ONE’s publication criteria as it currently stands. Therefore, we invite you to submit a revised version of the manuscript that addresses the points raised during the review process.

We look forward to receiving your revised manuscript.

Kind regards,

Tebelay Dilnessa, MSc

Academic Editor

PLOS ONE

Journal Requirements:

2. We note that your Data Availability Statement is currently as follows: [Yes - all data are fully available without restriction]

Additional Editor Comments (if provided):

The paper generally requires intensive revision. That is, it requires a through edition, revision and proofreading in terms of typographically, punctuation and grammatically.I would like to advise the author to structure the abstract as: Background, Objective, Method, and Conclusion.Add key words to the main document.In the abstract and result, the absolute number (numerator and denominator) is needed together with the percentage. For example, A/B (17.7%).

In the introduction part, the rationale is not explained well. Why you did this research? What has been done before, what is known before and which you intend to fill?

Statistical analysisAll ‘headings and subheadings’ should be bold.Please bring the following contents with headings above the reference lists. Acknowledgment, data availability statement, conflict of interest and author contributions.Figures are too poor and you have to prepare them again.The references were not written properly. All references should be written correctly according to Vancouver style.Use the alignment ‘Justify’ for the text throughout the manuscriptFollow the standard binomial nomenclature, italize journal name and the word ‘et al’Follow the guideline for manuscript writing protocol for PLoS One.

Reviewers' comments:

Reviewer's Responses to Questions

**Comments to the Author**

1. Is the manuscript technically sound, and do the data support the conclusions?

Reviewer #1: Partly

Reviewer #2: Partly

2. Has the statistical analysis been performed appropriately and rigorously?

Reviewer #1: No

Reviewer #2: No

3. Have the authors made all data underlying the findings in their manuscript fully available?

Reviewer #1: Yes

Reviewer #2: Yes

4. Is the manuscript presented in an intelligible fashion and written in standard English?

Reviewer #1: No

Reviewer #2: No

5. Review Comments to the Author

Reviewer #1: Information on mortality associated with diagnostic and treatment delay in extra-pulmonary tuberculosis is valuable data for the TB program. The manuscript needed major revision before it proceeded to the publications.

Major comments

Provide a clear description of what information is extracted from the previous studies and the information collected for this study. A precise detail is required on the data collection methods such as the source data and verification methods/quality control used.

Make sure that only information relevant to this study is included in the definition, and include a citation for each term.

The exclusion criteria are not clear. Certain patients such as those who failed to provide information on diagnostic or treatment delay and who started on ATT before inclusion in the study/diagnostic sampling, have been excluded. But the reason is not indicated or the mortality rate in this group has not been analyzed.

The table’s presentation is not easy to follow. Example. Table 1 adds a p-value and uses a separate column for unadjusted and adjusted OR. What does it mean for antiretroviral therapy coverage? Please check the number of people who do not receive anti-TB treatment (23-table 1 vs 21-text).

The description on health-seeking behavior is mainly focused on the health facilities visited. Consider a different terminology for health-seeking behavior and a concise presentation of key findings.

The result presentation under diagnostic and treatment delay is not easy to follow. Focus on the main findings.

Classifying cases as adenitis and non-adenitis may not be appropriate, while the data showed that most deaths occurred among pleuritis (10/39, 26%), peritonitis (8/39, 20%), and adenitis (6/39, 15%).

The key findings summary (lines 333–338) are remarkable. Besides, add other key findings like proportion of death by site of infection, health system diagnostic delay, and structure the discussion based on the key findings.

Minor comments

Line 27-29: In the abstract introduction, add the major knowledge gap beside the unavailability of data.

Line 30: Add how those 111 EPTB patients are diagnosed.

Line 59: have not analyzed association of diagnostic delay with mortality. Rephrase.

Line 78: does nationally refer to the global or Tanzania?

Line 80-89: Move to method section. Part of the information can be used to describe the study area.

Reviewer #2: line 32; all the patients should be included, not only those that survived

line 34, 35, what corresponding numbers are these? are they percent? and what measure of dispersion is being used? is it IQR or 95% CI?

LINE 39-41; This conclusion is not aligned with the aim stated. how was the diagnostic delay, how was treatment delay. then mention mortality although the authors must make it clear to the readers that they had limited power to make firm conclusions due to small numbers

paper needs review by an English editor for grammatical errors

line 80-89; is about study setting. this should be moved to methods section

line 95: Be clear that this is All cause mortality

Authors should include the study endpoints and how they intended to report them. currently they are not clear

line 146: i recommend the authors rethink the definition of diagnostic delay which include patients history. the recall and reporting bias is almost impossible to control in this approach and the results will have limited relevance to the readers. i recommend the authors limit the time from first contact with a health worker including at other health facilities

line 147: authors should justify why ATT delay includes sampling time. ATT should only be initiated where a diagnosis has been made. this measure will confuse the readers. please clarify

line 163-163: is not clear at all. Authors should clarify how this score is reached for interest of the readers

authors have made no mention of sample size derivation. this is paramount for this prospective cohort study

line 181-188: this section is not clear. Authors have not provided the measurement and outcome variables in the write up. so it is difficult to assess the appropriateness of the proposed analysis.

line 185: this approach is not clear while reviewing the results, both bivariate and multivariate is included in all the tables, not easy to know which was significant at bivariate or not. authors should revisit the analysis plan and follow it in presenting results

line 197: Table 1 is for for descriptive display. authors are advised to show the analysis in subsequent tables. this is otherwise confusing to the readers

i recommend the authors present results in chronology starting with the main outcomes to other factors.

line 269: there is no Box plot figure

line 292: Authors should maintain the same outcome measure otherwise the readers will get confused. let the outcome measures be clearly defined

line 332: i recommend authors to revisit the discussion and focus on the main aims of the study, explicitly state the findings and focus on significant findings on multivariate as associations

line 398-403: It is not clear whether the authors were stating the limitations of the study.

i recommend that the limitations of this study be clearly outlined for the readers

line 405: the conclusion does not align with the presented results. the authors should review and align

6. PLOS authors have the option to publish the peer review history of their article (what does this mean? ). If published, this will include your full peer review and any attached files.

**Do you want your identity to be public for this peer review?** For information about this choice, including consent withdrawal, please see our Privacy Policy .

Reviewer #1: No

Reviewer #2: No

---

## [Author Response · Author response to Decision Letter 1]

31 Dec 2024

Dear Sirs,

I would like to begin by expressing my sincere gratitude for the time and effort you have dedicated to reviewing our manuscript. Your valuable feedback has greatly contributed to enhancing the quality of the work, and we are confident that the revisions have strengthened it significantly.

We hope that, after these revisions, the manuscript will meet the standards for publication in PLoS ONE.

Below, you will find our detailed responses to the comments from both the editor and reviewers.

Thank you once again for your consideration and insightful suggestions.

Sincerely,

Erlend Grønningen,

M.D., DTMH

Response to editor:

The full dataset has been uploaded as a supporting information file (S2). The data availability statement has been revised to:

"The data required to replicate our results are included in the supporting information in Table S2."

Style Requirements:

• The abstract has been restructured according to the editor’s instructions.

• The file name for the supporting information now complies with PLoS ONE requirements.

Language Revisions:

A comprehensive edit, revision, and proofreading process has been completed to improve the clarity and quality of the manuscript.

Other Revisions:

• Keywords have been added to the main document.

• Absolute numbers have been included alongside percentages throughout the manuscript.

• The rationale in the introduction has been enhanced.

• Statistical analysis has been re-conducted following revisions to the exclusion criteria, and all tables have been updated accordingly.

• Headings and subheadings are now bolded.

• The acknowledgements, data availability statement, conflicts of interest, and authors' contributions have been moved above the references.

• References have been updated according to the journal's requirements and the editor’s comments.

• The text alignment has been set to ‘justify.’

• The writing protocol for PLoS ONE has been followed.

• Figures have been improved:

o Figure 1: Revisions were made to better highlight the different delays.

o Figure 2: The Y-axis of the box plots has been aligned, and the figure caption has been rewritten for clarity.

Response to reviewer one:

Comment: Provide a clear description of what information is extracted from the previous studies and the information collected for this study.

Response: This has been added to the methods in the manuscript.

Comment: A precise detail is required on the data collection methods such as the source data and verification methods/quality control used:

Response: This has been added to the methods section.

Comment: Make sure that only information relevant to this study is included in the definition, and include a citation for each term

Response: I have tried to reduce the text in the manuscript so that it better focuses on the research question in our paper: does diagnostic and treatment delay affect mortality in EPTB? Citations are entered for the different delays.

Comment: The exclusion criteria are not clear. Certain patients such as those who failed to provide information on diagnostic or treatment delay and who started on ATT before inclusion in the study/diagnostic sampling, have been excluded. But the reason is not indicated or the mortality rate in this group has not been analyzed.

Response: This was a very helpful comment. I have now explicitly stated the exclusion criteria in the manuscript. I see that they were not uniformly applied in the first version and the manuscript and have performed the statistical analyses again and updated the tables with the new number of patients. In general, this implies that all the patients that are included in our study now had information on diagnostic/treatment delay available. This reduces the samples size from 111 to 106.

For the analysis on delay in ATT initiation we chose to exclude those who were on ATT on inclusion. This was a deliberate choice as we did not want to have interference from empiric ATT.

We did also perform the diagnostic/treatment delay analysis including patients that were on ATT on inclusion, but the results did not differ from the results when this subset of patients was excluded. We chose not to include this analysis in the manuscript as it would introduce another sub-analysis that also did not show significant results.

Comment: The table’s presentation is not easy to follow. Example. Table 1 adds a p-value and uses a separate column for unadjusted and adjusted OR. What does it mean for antiretroviral therapy coverage? Please check the number of people who do not receive anti-TB treatment (23-table 1 vs 21-text).

Response: I have made separate rows for OR and aOR in table 1 and 2. For antiretroviral therapy coverage it shows that patients who were not on ART did not have higher mortality rates. The numbers have been aligned per the exclusion criteria above, and I again thank you for this comment.

Comment: The description on health-seeking behavior is mainly focused on the health facilities visited. Consider a different terminology for health-seeking behavior and a concise presentation of key findings.

Response: Changed heading of table to health facilities visited and removed data on diagnostic tests performed at first visit to remove redundant information from the manuscript.

Comment: The result presentation under diagnostic and treatment delay is not easy to follow. Focus on the main findings.

Response: This part of the manuscript has been rewritten, and the figures have been changed. The results on diagnostic/treatment delay are the key questions of this paper and has been moved above table 1-3 in the manuscript after comments from reviewer 2.

Comment: Classifying cases as adenitis and non-adenitis may not be appropriate, while the data showed that most deaths occurred among pleuritis (10/39, 26%), peritonitis (8/39, 20%), and adenitis (6/39, 15%).

Results: Again, this is a very good comment. One of the major limitations of this paper is the sample size. Before writing this paper, we performed the diagnostic/treatment delay and mortality analyses for all sites of infection. However, as the sample size for each site of infection was so small we ended up with extremely wide confidence intervals and results with too much uncertainty. In our previous papers we have also merged cases as adenitis and non-adenitis, while acknowledging that this represents a limitation. I have now stated this clearly as a limitation in the discussion.

Minor comments:

The minor changes have been amended in the revised manuscript.

Response to reviewer two:

Comments about abstract:

Have been amended in the revised manuscript

Comment: paper needs review by an English editor for grammatical errors

Response: English revision has been performed, and changes have been amended throughout manuscript.

Comment: line 80-89; is about study setting. this should be moved to methods section.

Response: Has been moved per comment

Comment: line 95: Be clear that this is All cause mortality

Response: Have specified this in the manuscript

Comment: Authors should include the study endpoints and how they intended to report them. currently they are not clear

Response: Have changed so that only one outcome measure is reported (mortality). This aligns with another of Your comments below.

Comment: line 146: i recommend the authors rethink the definition of diagnostic delay which include patients history. the recall and reporting bias is almost impossible to control in this approach and the results will have limited relevance to the readers. i recommend the authors limit the time from first contact with a health worker including at other health facilities

Response: Thank you for this comment. However, the separation of delays into patient and health systems delay is often how delays are reported (Storla et al, A systematic review of delay in the diagnosis and treatment of tuberculosis, BMC Public Health, 2008, Fetensa et al, Magnitude and determinants of delay in diagnosis of tuberculosis patients in Ethiopia: a systematic review and meta-analysis: 2020, Archives of Public Health, 2020). Recall and reporting bias does represent a limitation of our study, and after Your comments I have updated this under the limitations in the discussion.

Comment: line 147: authors should justify why ATT delay includes sampling time. ATT should only be initiated where a diagnosis has been made. this measure will confuse the readers. please clarify

Response: In the resource limited setting empiric ATT is widely used for EPTB. A change in clinical practice with reliance on diagnostic tools can also have harmful effects. Introduction of diagnostic tools that have imperfect sensitivities and longer processing times might cause harm if clinicians change their clinical practice. This was also raised as a major concern by some authors prior to the implementation of GeneXpert for PTB.

I have updated the manuscript with the following paragraph:

We included sample processing time in the analysis of ATT delay. The processing time potentially could have influenced delay in ATT initiation if clinicians had chosen to withhold ATT until they had a confirmed EPTB diagnosis.

Comment: line 163-163: is not clear at all. Authors should clarify how this score is reached for interest of the readers.

Response: I have moved the part on response to treatment next to the paragraph that describes the follow-ups. Have also added which objective findings that were assessed. The paragraphs have been merged:

Follow-up visits were arranged to assess the response to ATT at 2-3 months and at the end of treatment. For patients initially lost to follow-up, additional tracing was performed in January 2018. After the last patient was included in July 2017, the follow-up continued for an additional six months for the mortality analysis with data collection concluding in January 2018. Response to treatment was defined as a ≥3-point improvement in symptoms as measured by patient (I) or clinician (I), weight gain (I) and improvement of objective findings such as size of lymph nodes, pleural effusions and ascites (I. Response to treatment was assessed at both 2-3 months and 6 months.

Comment: authors have made no mention of sample size derivation. this is paramount for this prospective cohort study

Response: Have specified the following under methods:

The enrollment of patients and implementation of the MTP64 test were carried out regardless of sample size.

Have also specified in the last part of the discussion that a small sample size is a limitation of our study.

Comment: line 181-188: this section is not clear. Authors have not provided the measurement and outcome variables in the write up. so it is difficult to assess the appropriateness of the proposed analysis.

Response: Thank you. This was a very important clarification. I have updated the text as following:

For statistical analysis, Statistical Package for Social Sciences, version 28.0 (IBM, Armonk, NY), was used. To analyze factors associated with mortality, odds ratios (OR) for categorical variables were calculated using logistic regression. ORs were considered statistically significant at the 5% level if the 95% confidence intervals (CI) did not include 1.0. Adjusted odds ratios (aOR) were calculated by including all the variables that were significant in a multinominal logistic regression model, and statistical significance was defined in the same way as for ORs. For analyses on mortality and diagnostic/treatment delays, Mann-Whitney-test was applied for group comparisons, as the data were non-normally distributed. A p-value <0.05 was considered statistically significant.

Comment: line 185: this approach is not clear while reviewing the results, both bivariate and multivariate is included in all the tables, not easy to know which was significant at bivariate or not. authors should revisit the analysis plan and follow it in presenting results

Response: Thank you. Have separated the rows for OR and aOR in all the tables to make the reading of both the tables and analyses easier.

Comment: line 197: Table 1 is for descriptive display. authors are advised to show the analysis in subsequent tables. this is otherwise confusing to the readers

i recommend the authors present results in chronology starting with the main outcomes to other factors.

Response: This was a helpful comment. I have moved the order of the figures and table to better align with your comments.

Comment: line 269: there is no Box plot figure

Response: The figures were uploaded as separate files per instructions to authors. My apologies if the box plots did not reach you.

Comment: line 292: Authors should maintain the same outcome measure otherwise the readers will get confused. let the outcome measures be clearly defined

Response: Thank you, again an important point. Have removed table 3 and the analysis of ATT initiation with different outcome measures. I agree with the assessment of the reviewer that it is not aligned with the main aims of the study and that it is confusing.

Comment: line 332: i recommend authors to revisit the discussion and focus on the main aims of the study, explicitly state the findings and focus on significant findings on multivariate as associations

Response: These are good comments that focuses the paper on the main objectives. I have tried to focus the discussion, reduced the discussions of diagnostic tools and so forth

Comment: line 398-403: It is not clear whether the authors were stating the limitations of the study. i recommend that the limitations of this study be clearly outlined for the readers

Response: I have now outlined in the introduction to this paragraph that we are describing the limitations of the study.

Comment: line 405: the conclusion does not align with the presented results. the authors should review and align

Response: Has been changed accordingly

---

## [Decision Letter · Decision Letter 1]

21 Jan 2025

PONE-D-24-41844R1Diagnostic and treatment delay in extrapulmonary tuberculosis and association with mortality: experiences from Mbeya, Tanzania.PLOS ONE

Dear Dr. Grønningen,

Thank you for submitting your manuscript to PLOS ONE. After careful consideration, we feel that it has merit but does not fully meet PLOS ONE’s publication criteria as it currently stands. Therefore, we invite you to submit a revised version of the manuscript that addresses the points raised during the review process.

We look forward to receiving your revised manuscript.

Kind regards,

Tebelay Dilnessa, MSc

Academic Editor

PLOS ONE

Journal Requirements:

Additional Editor Comments:

It needs a through proofreading.Line 72: EPTB; a sentence cannot start by abbreviation, write its full formLine 181: The description of Fig 1should be described in terms of person, place and time.Line 201: Previously I commented to make a correction by saying, ‘Statistical analysis’. Do you have any justification?

Reviewers' comments:

Reviewer's Responses to Questions

**Comments to the Author**

1. If the authors have adequately addressed your comments raised in a previous round of review and you feel that this manuscript is now acceptable for publication, you may indicate that here to bypass the “Comments to the Author” section, enter your conflict of interest statement in the “Confidential to Editor” section, and submit your "Accept" recommendation.

Reviewer #1: (No Response)

Reviewer #3: All comments have been addressed

2. Is the manuscript technically sound, and do the data support the conclusions?

Reviewer #1: Yes

Reviewer #3: Yes

3. Has the statistical analysis been performed appropriately and rigorously?

Reviewer #1: Yes

Reviewer #3: Yes

4. Have the authors made all data underlying the findings in their manuscript fully available?

Reviewer #1: Yes

Reviewer #3: Yes

5. Is the manuscript presented in an intelligible fashion and written in standard English?

Reviewer #1: Yes

Reviewer #3: Yes

6. Review Comments to the Author

Reviewer #1: The manuscript is revised well. Only a few points are left unaddressed.

The description of the population and study design is extensive. It would be great to structure the information under subheadings such as enrollment, data collection, etc. Also, make sure the data source for the current study is clearly described.

Table 2: The category under referral to this facility is three, but the corresponding number is four. Please check.

Reviewer #3: No further comments. The authors have addressed all the comments.

I think the manuscript is now fit for publication.

7. PLOS authors have the option to publish the peer review history of their article (what does this mean? ). If published, this will include your full peer review and any attached files.

**Do you want your identity to be public for this peer review?** For information about this choice, including consent withdrawal, please see our Privacy Policy .

Reviewer #1: No

Reviewer #3: No

---

## [Author Response · Author response to Decision Letter 2]

20 Feb 2025

Dear Sirs,

Thank You for your time and consideration in reviewing our manuscript (ONE-D-24-41844R1) titled "Diagnostic and treatment delay in extrapulmonary tuberculosis and association with mortality: experiences from Mbeya, Tanzania."

We appreciate the valuable feedback provided by the reviewers and the editorial team. In this revised version, we have addressed all the points raised and made the necessary modifications to enhance the clarity, accuracy, and readability of our manuscript.

Below, we provide a point-by-point response to the comments from the editor and reviewers:

Editor’s Comments:

1. Proofreading:

o A thorough proofreading of the manuscript has been conducted to correct typographical, grammatical, and clarity issues.

2. Line 72: EPTB abbreviation

o The full form "Extrapulmonary Tuberculosis (EPTB)" has been written out at the beginning of the sentence.

3. Line 181: Description of Fig 1 should include person, place, and time.

o The following descriptions have been added to ensure clarity:Patient delay: The time (days) from the patient’s report of symptom onset to the first visit to a healthcare provider.Healthcare system diagnostic delay: The time from the first healthcare visit to diagnostic sampling for suspected EPTB. Total diagnostic delay: The sum of patient delay and healthcare system diagnostic delay. Delay in ATT initiation: The time from diagnostic sampling to the start of anti-tuberculosis treatment. Total treatment delay: The time from symptom onset to the start of ATT.

4. Line 201: Correction to "Statistical analysis"

o Apologies for the oversight. The correction has been made as per the suggestion.

5. Reference Check for Retractions:

o The citation of the Global Tuberculosis Report has been updated to the 2024 edition.

o A comprehensive review of the reference list was conducted. No cited articles were found to be retracted. A manual search in PubMed and an AI-based search for retraction notifications confirmed this.

Reviewer #1’s Comments:

1. Structuring Population and Study Design Section:

o The description of the study design and population has been restructured under subheadings (e.g., Exclusion criteria, Data sources) to improve readability.

o The data source for the study is now clearly stated within this section.

2. Table 2: Clarification on Referral Categories:

o The visual structure of Table 2 has been revised to clearly indicate that there are four referral categories. Additional table lines were added to avoid confusion regarding the number of categories.

Reviewer #3’s Comments:

• Reviewer #3 has indicated that all comments have been satisfactorily addressed, and no further revisions were required.

We sincerely appreciate the time and effort taken by the reviewers and editorial team to provide constructive feedback. We believe that the revisions have strengthened the manuscript and made it suitable for publication in PLOS ONE.

Thank you for your consideration. We look forward to your feedback.

Best regards,

Erlend Grønningen

MD, PhD, DTMH

---

## [Editor Report · Decision Letter 2]

23 Feb 2025

Diagnostic and treatment delay in extrapulmonary tuberculosis and association with mortality: experiences from Mbeya, Tanzania.

PONE-D-24-41844R2

Dear Dr. Grønningen,

We’re pleased to inform you that your manuscript has been judged scientifically suitable for publication and will be formally accepted for publication once it meets all outstanding technical requirements.

Kind regards,

Tebelay Dilnessa, MSc

Academic Editor

PLOS ONE
---

## [Editor Report · Acceptance letter]

PONE-D-24-41844R2

PLOS ONE

Dear Dr. Grønningen,

I'm pleased to inform you that your manuscript has been deemed suitable for publication in PLOS ONE. Congratulations! Your manuscript is now being handed over to our production team.

Kind regards,

on behalf of

Dr. Tebelay Dilnessa

Academic Editor

PLOS ONE